# Effect of Lyoprotective Agents on the Preservation of Survival of a *Bacillus cereus* Strain PBG in the Freeze-Drying Process

**DOI:** 10.3390/microorganisms11112705

**Published:** 2023-11-04

**Authors:** Diana Galeska Farfan Pajuelo, Milena Carpio Mamani, Gisela July Maraza Choque, Dina Mayumi Chachaque Callo, César Julio Cáceda Quiroz

**Affiliations:** Bioremediation Laboratory, Jorge Basadre Grohmann National University, Tacna 230001, Peru; dfarfanp@unjbg.edu.pe (D.G.F.P.); mcarpiom@unjbg.edu.pe (M.C.M.); gmarazac@unjbg.edu.pe (G.J.M.C.); dmchachaquec@unjbg.edu.pe (D.M.C.C.)

**Keywords:** lyophilization, bacterial survival, mannitol, glucose, lactose and *Bacillus*

## Abstract

Lyophilization is a widely employed long-term preservation method in which the bacterial survival rate largely depends on the cryoprotectant used. *Bacillus cereus* strain PBC was selected for its ability to thrive in environments contaminated with arsenic, lead, and cadmium, tolerate 500 ppm of free cyanide, and the presence of genes such as *ars*, *cad*, *ppa*, *dap*, among others, associated with the bioremediation of toxic compounds and enterotoxins (*nheA*, *nheB*, *nheC*). Following lyophilization, the survival rates for Mannitol 2.5%, Mannitol 10%, and Glucose 1% were 98.02%, 97.12%, and 96.30%, respectively, with the rates being lower than 95% for other sugars. However, during storage, for the same sugars, the survival rates were 78.71%, 97.12%, and 99.97%, respectively. In the cake morphology, it was found that the lyophilized morphology showed no relationship with bacterial survival rate. The best cryoprotectant for the PBC strain was 1% glucose since it maintained constant and elevated bacterial growth rates during storage, ensuring that the unique characteristics of the bacterium were preserved over time. These findings hold significant implications for research as they report a new *Bacillus cereus* strain with the potential to be utilized in bioremediation processes.

## 1. Introduction

In recent years, mining has raised significant environmental and public health concerns [1,2,3,4], resulting in the degradation of vast land areas worldwide [5], impacting land [6], water [7,8], soil [9,10], and air quality [11,12,13]. The situation is exacerbated when mining waste deposits, considered as environmental mining legacies, are left abandoned or inactive [14]. Globally, there are over a million abandoned mines [15,16,17], and Peru is no exception to this issue. In the Tacna region, mining legacies from the Chulluncane copper mine contain high levels of contamination, with 1102 mg/kg of arsenic, 620.33 mg/kg of lead, 27.83 mg/kg of cadmium, and 0.04 mg/kg of free cyanide (CN^−^) [18].

Faced with this ongoing threat, the scientific community has developed strategies for the bioremediation of contaminated environments, with a particular focus on microorganisms [19]. Microorganisms are considered promising due to their metabolic activity [20,21,22], adaptability to environmental stress [23,24], and the effectiveness of indigenous bacteria in bioremediation processes [25,26].

The *Bacillus* genus has emerged as a valuable tool in remediating various toxic metals. Notable species such as *B. subtilis*, *B. cereus*, and *B. thuringiensis* have been extensively studied for their potential in this field [27]. Additionally, other species such as *B. sterothemophilus*, *B. megaterium*, *B. pumilus*, *B. licheniformis*, and *B. jeotgalim* have shown bioremediation capabilities [28,29,30,31], producing various secondary metabolites such as lipopeptides, polypeptides, macrolides, fatty acids, polyketides, and isocoumarins, which enable them to decontaminate soils polluted with toxic metals and organic pollutants [32].

*Bacillus cereus*, in particular, has been employed in various studies to break down toxic metals [33,34,35], hydrocarbons [36,37], free cyanide [38,39], pesticides [40,41], and polypropylene [42]. Given its broad applicability in bioremediation processes [43,44], unique biotechnological properties [20,21,22], involvement in the microbiota of contaminated soils, and tolerance for free cyanide, it is crucial to address the preservation of *Bacillus cereus* [45]. This ensures the conservation of the biological characteristics of the dehydrated microorganism [19,46] and prevents the loss of its metabolic activity [19]. This preservation is achieved by applying an appropriate protective agent during lyophilization and creating optimal conditions for this process.

Lyophilization, a long-term preservation technique, holds significant importance in scientific research and industry [43,44]. It aims to conserve Microorganisms [45], preserving their microbial cells under pure and homogeneous conditions, ensuring their stability and maintaining their morphological, physiological, and genetic traits for decades [46,47,48]. Successful preservation is achieved through the prior freezing of water within the material, effectively halting chemical, biochemical, and microbiological processes. As a result, microbial cells maintain their integrity and homogeneity, ensuring stability over decades, exceeding a ten-year period, all while preserving their morphological, physiological, and genetic characteristics [47,48].

However, lyophilization does lead to an inevitable reduction in functional activities [49,50,51], cell damage, and loss of viability [44,52], causing mechanical damage, damage from the effects of the solution, changes in membrane integrity [48,53], protein denaturation, alterations in pH dynamics [44], and the formation of intracellular-extracellular ice crystals due to the environmental stress produced by freezing and drying (sublimation-desorption) [54].

To prevent the loss of viability, it is necessary to improve conditions during lyophilization, including the proper use of different lyoprotectors, evaluation of cell damage mechanisms, lyophilization conditions, and optimization of parameters (temperature, pressure, and time). Among these, lyoprotectors are considered the most critical, as they affect cell viability during the lyophilization process [44,55,56], aiming to reduce cell damage during both the process and rehydration [44], while maintaining physiological activity and storage stability during long-term preservation [57].

Commonly used lyoprotectors include disaccharides. The most frequently used ones are 1% and 5% glucose, 4.74%, 5%, 7.5%, and 10% trehalose [43,58], 1.45% xylitol, 5% cellobiose, 5% d-galactose, 5%, 7%, and 10% sucrose [45,57], 5% fructose, and 7% lactose [59].

However, the mechanisms of lyoprotector protection are not yet fully understood and generally involve preventing ice formation both inside and outside cells, replacing hydrogen bonds in water, or creating a glassy matrix [45]. Consequently, there is no universally applicable approach to the successful preservation of all bacteria [50,60], as each protective agent uniquely affects microorganisms, and a universal protection formulation has yet to be devised [61,62,63,64].

Therefore, it is essential to evaluate lyoprotectors and appropriate suspension media because their effectiveness varies depending on the microorganism you intend to preserve [50,54]. In this context, the objective of this study was to assess the effectiveness of different cryoprotectant agents (such as mannitol, glucose, and lactose) at various concentrations to achieve the highest survival rate of the *Bacillus cereus* PBG strain during the lyophilization process and subsequent storage. This is aimed at ensuring that the microorganism retains its inherent characteristics acquired prior to lyophilization. Additionally, we aim to investigate the potential relationship between the cake morphology obtained during lyophilization and bacterial viability. These findings will contribute to the identification of the most suitable cryoprotectant for the long-term preservation of this bacterial strain, thereby facilitating its future application in the bioremediation of contaminated environments.

## 2. Materials and Methods

### 2.1. Sample Collection, Bacterial Isolation

In this study, soil samples were collected from abandoned mining areas in the Palca district, Tacna, Peru (81°13′94.0″ N, 03°58′60.2″ E), at a depth of 15 cm, which were contaminated with arsenic, lead and cadmium. To isolate the bacterium, 100 g of soil was mixed with 200 mL of sterile distilled water, manually homogenized for 1 h, and allowed to settle for 5 min to obtain the supernatant [65]. To enrich and isolate the bacteria, 10 mL of the resulting liquid were then transferred to 90 mL of nutrient broth, which was incubated at 30 °C for 24 h at 150 rpm. For adaptation to alkaline conditions, 10 mL of the supernatant was inoculated into 90 mL of nutrient broth adjusted to pH 10.5, and incubated at 30 °C at 150 rpm for 24 h. Subsequently, the samples were plated on nutrient agar plates from Merck Company (Darmstadt, Germany) and incubated at 35 °C for 24 h [66].

### 2.2. Strain PBG Tolerance Test to CN^−^

The isolated bacterial strains were subjected to tests to assess their resistance to CN^−^. To conduct these tests, 10 mL of the enriched medium was transferred to 90 mL of a liquid mineral medium 9M composed of (g/L): Na_2_HPO_4_.7H_2_O (12.8); KH_2_PO_4_ (3); NaCl (0.5); MgSO_4_.7H_2_O (0.5); CaCl_2_ (0.1); sodium acetate at 0.2% (*w*/*v*) as a carbon source, yeast extract at 0.2% (*w*/*v*) as a nitrogen source; and 1% (*v*/*v*) of a mineral salts solution containing (g/L): ZnSO_4_.7H_2_O (0.05); MnCl_2_.4H_2_O (0.05); CuCl_4_.2H_2_O (0.005); Na_2_MoO_4_.2H_2_O (0.005); Na_2_B_4_O_7_.10H_2_O (0.002); CoCl_2_.6H_2_O (0.0003). The pH was adjusted to 10.5, as described by Huertas et al., (2010) [67] and Khamar et al., (2015) [66].

To determine their tolerance to CN^−^, the strains were inoculated into the enriched 9M medium with yeast extract, to which different concentrations of cyanide, ranging from 50 to 500 ppm, were added, with the pH adjusted to 10.5 to prevent cyanide volatilization. Subsequently, the samples were incubated at 30 °C for 24 h. After incubation, the viability of the isolated strains at each CN^−^ concentration was assessed by culturing on nutrient agar plates.

A representative colony of each of the isolated strains was preserved on nutrient agar at 4 °C and labeled as P_02, P_06, P_07, P_11, P_12, P_13, P_14, P_16, P_23, and P_24, respectively. Finally, the strain with the highest cyanide tolerance was selected for lyophilization. Additionally, this strain was chosen due to its origin in a toxic metal-contaminated environment and was labeled as PBG.

### 2.3. Microorganism and Microbial Culture

PBG was cultured in trypticase soy broth (TSB) and incubated at 30 °C for 24 h to establish the bacterial growth curve and define the calibration curve (which relates the absorbance to the CFU/mL). A 2% inoculum was used in 150 mL of TSB and incubated at 30 °C for 12 h using an orbital shaker at 150 rpm. To determine the bacterial concentration, hourly measurements were performed using an EPOCH spectrophotometer (BioTek, Winooski, VT, USA) at a wavelength of 600 nm and by plate count (CFU/mL).

### 2.4. Sequencing and Molecular Identification

Genomic DNA was obtained from the strain PBG; for this purpose, the bacteria were cultured on Luria Bertoni agar and incubated at 35 °C for 24 h. Subsequently, the ImmunoPREP bacterial DNA extraction kit (Analytik Jena, Jena, Germany) was used, followed by DNA quantification using a Qubit 4 fluorometer (Life Technologies, Carlsbad, CA, USA) and quality evaluation using a 1% agarose gel. Then, a paired-end library was constructed using an Illumina Prep DNA library preparation kit (Illumina, Cambridge, UK). Unique dual indexes from the Nextera DNA CD Indexes kit (Illumina, Cambridge, UK). Finally, libraries were sequenced as 2 × 151 bp paired-end reads on the Illumina Miseq platform using a 600-cycle reagent kit. Finally, the quality of the sequences was evaluated with the fastQC program (v0.11.9), Quality trimming (Q > 25) and adapters were removed with the program Trimmomatic (v0.39). Assembly was performed with the program SPAdes (v.3.15.4) and the 16S rRNA gene sequence was obtained with RNAmmer (v1.2), then a comparative identity search was performed using the BLAST tool (Basic Local Alignment Search Tool).

### 2.5. Selection of Lyoprotectants and Lyophilization Conditions

Three types of lyoprotectants that have been reported to be common for this bacterium and at different concentrations were selected for glucose (1, 5 and 10%), mannitol (2.5, 5 and 10%) and lactose (5, 7.5 and 10%). The lyoprotectant media were suspended in distilled water that was sterilized using 0.22 µm Millipore Express* PES Membrane Filter Unit syringe filters [43,54,68].

To perform the lyophilization process, bacterial cells in the stationary phase were used (CFU/mL) and then centrifuged at 12,000 rpm for 12 min. The pellet was then resuspended in 3 mL of lyoprotectant using 10 mL vials, and the sample was frozen at −80 °C for 48 h. The samples were then dried in a freeze dryer (FREEZONE 4.5 L-LABCONCO) at −80 °C for 24 h at 0.000 mBar pressure.

### 2.6. Rehydration and Cell Viability Determination

The lyophilized samples were resuspended in 3 mL of TSB and incubated at 30 °C for 24 h. Subsequently, the bacterial survival rate was determined after lyophilization and every 19 days during storage at 4 °C for a period of 76 days. The survival rate was calculated using the formula described by Peiren et al., (2016) [43]:(1)$$Survival\left(\%\right)=\frac{\left(Log10\;(CFU/mL)\;after\;lyophilization\right)}{\left(Log10\;(CFU/mL)\;before\;lyophilization\right)}\times 100$$

### 2.7. Cake Morphology

For the evaluation of the morphology of freeze-dried cultures, 21 vials were used for each lyoprotectant. Visual classification was performed following the method described by Peiren et al., (2016) [43], wherein the classification criteria included: (i) intact, indicating the absence of cake contraction; (ii) porous, characterized by the presence of large holes in the cake without evident contraction; (iii) partial collapse, marked by observable contraction along the cake is perimeter, although its height remained unaffected; and (iv) collapse, where contraction along the cake is perimeter was evident as a decrease in its height, with or without melting.

### 2.8. Statistical Analysis

Statistical analysis was performed using RStudio version 4.1.2. One-way analysis of variance (ANOVA) was applied to evaluate the significant differences between the means of the different treatments (*p* < 0.05). Duncan’s test was used to compare the groups with the control after the freeze-drying process. In addition, Fisher’s test was used to compare all study groups during storage. In case the data did not meet the assumptions of normality, the Johnson data transformation was applied before assessing significant differences.

## 3. Results

### 3.1. General Characteristics of Strain PBG

A total of 24 bacterial strains were isolated from samples collected from environmental residues at the abandoned Chullancane mine, which exhibited a legacy contamination of Arsenic, Lead, and Cadmium, and the presence of Barium, Cadmium, Chromium, and free cyanide. Among these isolates, 10 strains demonstrated cyanide tolerance ranging from 100 to 500 ppm (Figure 1), with strain P_07 displaying tolerance up to 500 ppm.

The PBG strain is a Gram-positive bacterium that produces rod-shaped endospores and forms irregular creamy colonies on tryptic soy agar (TSA). These colonies exhibited a morphology characterized by a shiny and frosted surface, creamy and smooth texture, superficial localization, notably raised elevation, irregular shape, undulated edge, and a cream color tone.

The data presented in Table 1 indicate that the isolated strain is taxonomically attributable to the genus *Bacillus.*

### 3.2. Sequencing Findings

The 16S gene of the PBG strain was identified using the BLAST alignment tool and compared with nucleotide sequences available in the GenBank database. The sequence had a length of 1540 base pairs and exhibited a similarity of over 99% with *Bacillus cereus* (sequence identifiers: CP053931.1, CP053997.1, CP053991.1), *B. thuringiensis* (sequence identifiers: CP054568.1, CP053938.1, CP053934.1), and *B. anthracis* (sequence identifiers: CP054816.1, CP054800.1, CP054797.1). Furthermore, the results of biochemical tests revealed significant similarities with the PBG strain, indicating a close molecular and genetic relationship among these bacteria, thereby complicating the precise identification of the species. This can be attributed to the high similarity in the 16S rRNA gene sequences, which range between 92% and 100% [69], rendering them virtually indistinguishable from a phylogenetic perspective [70]. However, a more detailed molecular characterization was conducted, including a phylogenetic reconstruction using the maximum likelihood algorithm, which placed the PBG strain within the B. cereus clade. Another significant finding was the detection of enterotoxin genes nheA, nheB, and nheC, Cytk, along with genes associated with toxic metal tolerance and those involved in cyanide degradation (refer to Table 2). On the other hand, genes encoding Cry proteins were not found [71], excluding *Bacillus thuringiensis*. Genes pagA, lef, and cya were also absent [72], eliminating the possibility of *B. anthracis*. Therefore, based on a more comprehensive molecular characterization that encompasses the identification of specific genes, taxonomic analysis, and microscopic features, such as the presence of Gram-positive bacilli forming endospores and the biochemical properties of the PBC strain, we can conclusively affirm its classification within the species *Bacillus cereus*.

### 3.3. Microbial Growth Characteristics of Bacillus cereus Strain PBG

In the evaluation of the growth curve, it was observed that the latency phase had a duration of three hours, then the logarithmic phase was approximately between 3 and 7 h, followed by the stationary phase from the seventh hour. So, the lyophilization process was performed at the beginning of this stage because at that time, it reached an average cell concentration of 6.05 × 10^10^ CFU/mL.

It is important to highlight that bacteria exhibit sustained development and are confronted with various fluctuations, which induce stress responses and contribute to the survival of the microbial population [46,49,73,74] (Figure 2).

The specific growth velocity for *Bacillus cereus* strain PBG was 1.34 h^−1^.

Likewise, the calibration curve CFU/mL vs. absorbance (OD 600) was calculated to determine the concentration of CFU/mL per unit of absorbance using the following equation:Y_CFU/mL_ = 4.7 × 10^10^(X_abs_) − 3.3 × 10^9^(2)

This predictive model showed a high statistical significance with a *p* value < 0.001 and a confidence interval of 95% (Figure 3).

### 3.4. Survival after Lyophilization and during Storage

The survival rate of *Bacillus cereus* strain PBG without the addition of any lyoprotectant (control group) was 72.16%. During the 76 days of evaluation, no significant differences in bacterial survival were observed between the control group and the 1%, 5% and 10% glucose and 10% lactose treatments (Figure 4). However, the 1% glucose treatment showed a higher survival rate, reaching 96.30%. On the other hand, treatments with 2.5%, 5% and 10% mannitol and 5% and 7.5% lactose showed significant differences in bacterial survival. These results indicate that the choice of lyoprotectants directly influences the observed survival rate.

When analyzing the impact of lyoprotective agents in the freeze-drying process, it was observed that glucose showed a tendency to maintain stable survival percentages. In particular, it was noted that 1% glucose achieved survival rates higher than 96.30%. On the other hand, it was found that the lyoprotectant lactose tended to increase the survival percentage, in contrast to the effect observed for mannitol, which showed a decrease in survival (Table 3).

The survival rate for 2.5% mannitol was the best after lyophilization, with a value of 98.02%. However, it tended to decrease over time, reaching a survival rate of 78.71%, similar to 10% mannitol, which decreased from 97.12% to 79.21% during the days evaluated (Figure 5).

### 3.5. Variation in the Cake Morphology

The variation in the structure of the cake (Figure 6), in all of the evaluations carried out, showed that mannitol presented high percentages of intact aspect in 70% (Figure 7), while in the case of glucose and lactose, the porous aspect prevailed in their different concentrations.

## 4. Discussion

Globally, mining activity has resulted in the degradation of approximately one billion hectares of land [5] and the persistent accumulation of hazardous waste [2] over time [75]. To address this challenge, the widely recognized strategy for eliminating such contaminants and restoring ecosystems is the application of microorganisms with potential in bioremediation processes. Particularly noteworthy is the essential role of indigenous microorganisms, which excel in their ability to generate various metabolites and their resilience to extreme conditions. This combination of attributes makes them an effective tool for optimizing the efficiency of bioremediation processes [25,26,76]. In this context, current perspectives in bioremediation are focused on identifying new microorganisms from contaminated environments [77].

The PBG strain stands out due to several outstanding features. First, it originates from environments affected by mining activities related to copper extraction, resulting in the presence of contaminants such as arsenic, lead, cadmium, barium, chromium, mercury, and free cyanide [18]. Secondly, it demonstrates remarkable tolerance to concentrations of up to 500 ppm of cyanide (CN^−^). Thirdly, genes associated with endotoxins, mechanisms of resistance to toxic metals, and cyanide have been identified, all of which are attributes that can be harnessed in environmental decontamination applications. Given the relevance of these traits and considering the widespread use of the *Bacillus* genus in bioremediation processes, the focus was placed on improving the lyophilization process. This is essential since the use of an appropriate cryoprotectant is critical to ensure bacterial survival, as its effectiveness may vary depending on the bacterial species [78]. It involves cultivating organisms under conditions that enhance their lyophilization tolerance, along with collecting them at the right time and optimizing the cryoprotectant [49]. Additionally, it is important to note that the choice of these agents may vary depending on the bacterial species [78].

The identification of *Bacillus cereus* strain PBG became a complex process because it belongs to a group known as *Bacillus cereus sensu lato*. These bacteria share a high genetic similarity, making precise species-level identification challenging, as they share 16S rRNA gene sequences with nucleotide similarity ranging from 92% to 100% [69]. To overcome this challenge, a more detailed molecular characterization was conducted, including phylogenetic reconstruction and the search for specific genes (Table 2), in combination with biochemical tests and microscopic and macroscopic observations. This analysis conclusively confirmed that strain PBG belongs to the *Bacillus cereus* species.

It is essential to highlight that the PBG strain exhibits unique functional attributes, such as the detection of significant genes, including enterotoxins *nheA*, *nheB*, and *nheC*, *CytK* [79,80,81]. Additionally, genes related to toxic metal tolerance and genes involved in cyanide tolerance were identified. These findings further support the uniqueness and importance of the PBG strain in biotechnology and bioremediation-related applications and studies.

According to our findings, the growth stationary phase in *Bacillus cereus* strain PBG was initiated at approximately the seventh hour of culture in TSB. However, previous research conducted by Shu et al., (2018) and Han et al., (2018) on *Lactobacillus acidophilus* using MRS broth (De Man, Rogosa, and Sharpe) as a culture medium and on *Bacillus amyloliquefaciens* B1408 using Luria-Bertani medium showed entry into the growth stationary phase between 36 and 48 h. This variation in timing could be attributed to differences in research methods, bacterial strains used, and specific culturing conditions in each case [82,83].

It is essential to emphasize that when bacteria are in the growth stationary phase, they exhibit a greater capacity for survival during the lyophilization process, as their morphological and physiological characteristics remain stable, ensuring consistent growth [44]. This finding aligns with previous research that suggests the optimal time for lyophilization of bacteria is at the end of the logarithmic phase and the beginning of the stationary phase. This is because, at this point, bacteria display increased resistance to adverse conditions such as desiccation, hyperosmolarity, and variations in pH and temperature [44,46,49,84,85,86]. Furthermore, in stages beyond the stationary phase, there is depletion of essential nutrients and bioelements [46], triggering stress responses in bacteria, including decreased cell volume, nucleoid compaction, changes in membrane composition, and cell wall structure [87,88].

Based on our experience, we recommend using TSB medium for the cultivation of *Bacillus cereus* strain PBG, as it facilitates entry into the stationary phase in a shorter period of time. This results in a mean initial microbial concentration of 6.05 × 10^10^ CFU/mL, which falls within the range suggested by Palmfeldt et al., (2003), who established an initial concentration of 10^7^–10^11^ CFU/mL. These favorable conditions positively influence higher survival rates and optimal cryoprotector distribution within the intracellular space [44,49,74], indicating that higher initial cell concentration prolongs the survival of viable cells. This is crucial as most cells tend to die during long-term storage [46,51].

In the field of biotechnology, various strains of *Bacillus cereus* have been lyophilized to harness their capabilities. The survival rate in lyophilization can vary depending on the type of cryoprotectant used. In the course of this research, different cryoprotectants were evaluated, and it was observed that, after lyophilization, the most effective ones were 2.5% mannitol, 10% mannitol, and 1% glucose, with survival rates of 98.02%, 97.12%, and 96.30%, respectively. These values significantly outperformed those recorded in the control group, which reported a survival rate of 72.16%. The protective effect of mannitol has been attributed to its ability to form crystalline structures that would protect the cell’s functional proteins [89], providing mechanical resistance [51] and replacing water extracted from the lipid membranes [49]. In contrast, glucose replaces water through hydrogen bonds in bacterial membrane bilayers [49] and forms amorphous glass matrices that minimize molecular mobility both inside and around the cell [46,73].

This high survival rate was attributed to the sporulated nature of *Bacillus cereus* (PBG strain), which conferred greater resistance to lyophilization compared to non-sporulated bacteria [82]. The survival rates observed in this study significantly exceeded those recorded for non-sporulated strains, such as *Bifidobacterium infantis*, which exhibited a survival rate of 86% using a 5% cellobiose, and the rates obtained for *Lactobacillus acidophilus* and *Lactobacillus salivarius*, as reported in the studies by Basholli et al. (2014) and Zayed & Roos (2004) [68,90]. Additionally, it is important to emphasize the influence of the rehydration medium used in the lyophilization process (TSB), as a complex medium may have the capacity to repair damaged cells [91].

Strains of *Bacillus cereus* have been subjected to lyophilization processes to harness their potential as biocontrol agents against pathogens and for their effectiveness in mercury removal. These microorganisms have undergone lyophilization to stabilize the cells for future research, focusing on possible scaling-up applications and formulation development, as suggested in the studies by Bhattacharya et al. (2016) and Bhattacharya et al. (2014). In most cases, processes are evaluated post-lyophilization. However, to ensure the long-term success of this process, it is essential to conduct assessments over time [92,93].

During the storage period, a significant decrease in survival rates was observed for mannitol at concentrations of 2.5% and 10% over the 76 days of evaluation. This decrease could be attributed to the formation of stable crystals that may cause damage to cell membranes [94], resulting in a lack of stability in membrane bilayers [24], protein destabilization [51], and the absence of amorphous crystal formation during storage [94]. In contrast, the survival rate remained constant for 1% glucose, attributed to this monosaccharide’s ability to form an amorphous glass matrix [51]. This matrix preserves components produced by cells before and during storage, preventing irreversible electrochemical changes in the cell membrane [52,74]. These results are consistent with those obtained by Han et al. (2018), who found that 1% glucose was the most effective monosaccharide for the survival of *Bacillus amyloliquefaciens* B1408, with rates exceeding 52%. However, the 10% glucose cryoprotectant showed significant decreases in survival, with reductions of up to 7.60, 8.84, and 9.46 logarithmic units for *Lactobacillus rhamnosus* CTC1679. For 7% lactose, decreases below 1 logarithmic unit were observed. This indicates that there is no universal cryoprotectant, and it depends on the microorganism [95].

The analysis of cake morphology after the lyophilization process revealed that the mannitol cryoprotector had an intact appearance within a range that varied from 86% to 100%. These results were due to mannitol, a sugar alcohol that provides mechanical resistance to the lyophilized cake, giving it an “elegant” appearance [96]. Although both glucose and lactose are also used as bulking agents to provide an appropriate structure to lyophilized cakes, they exhibited a higher percentage of porous appearance, exceeding 60%. In general, the morphology did not show a direct relationship with the survival rate. It is important to note that in an industrial context, results based on morphology might be considered unfavorable. However, according to Patel et al. (2017), a porous appearance does not imply a negative impact on protein stability [97].

Storage conditions have a critical impact on the viability of lyophilized cells, as highlighted in previous research [98]. Therefore, it is imperative to continue advancing research efforts to refine the selection of the most effective cryoprotectant and determine the optimal lyophilization conditions that ensure long-term benefits. Moreover, it is essential to investigate whether all genetic characteristics are truly preserved after the lyophilization process. The *Bacillus cereus* PBG strain proves to be versatile and offers a potential solution for the decontamination of environments affected by specific genes.

The advancements achieved in this research lay a solid foundation for the development of a viable alternative in the lyophilization of bacteria capable of surviving in extreme environments and withstanding the presence of cyanide. This strain, through bioaugmentation, could provide an effective solution for the decontamination of environments affected by the presence of cyanide and toxic metals, as well as in other bioremediation processes.

## 5. Conclusions

*Bacillus cereus* PBG is a novel bacterial strain isolated from environmental mining residues of an abandoned mine contaminated with arsenic, lead, and cadmium. It exhibits the capability to tolerate free cyanide, and molecular identification using 16S rRNA gene sequence analysis posed a challenge. However, a more detailed molecular analysis and phylogenetic reconstruction classified strain PBG as *Bacillus cereus*. Further molecular characterization revealed genes associated with the bioremediation of toxic compounds. This new strain demonstrated the ability to tolerate 500 ppm of CN^−^, rendering it even more suitable for lyophilization.

A contributing factor to the success of lyophilization was the use of an appropriate culture medium, such as trypticase soy broth (TSB), which facilitated optimal bacterial growth, enhancing survival rates and ensuring an effective distribution of the cryoprotectant.

For the novel *Bacillus cereus* PBG strain, the most effective cryoprotectant was 1% glucose, significantly improving bacterial survival rates at 96.30% after lyophilization. Furthermore, these rates remained constant during the 76 days of storage, ensuring prolonged bacterial survival.

The porous appearance of the lyophilized product using 1% glucose did not exhibit a direct correlation with bacterial survival rates, indicating no adverse impact on protein stability.

These findings bear significant implications for research and industry. They underscore the importance of carefully selecting cryoprotectants and culture media during lyophilization and emphasize the value of seeking novel strains with bioremediation potential for their possible biotechnological applications.

## Figures and Tables

**Figure 1 microorganisms-11-02705-f001:**
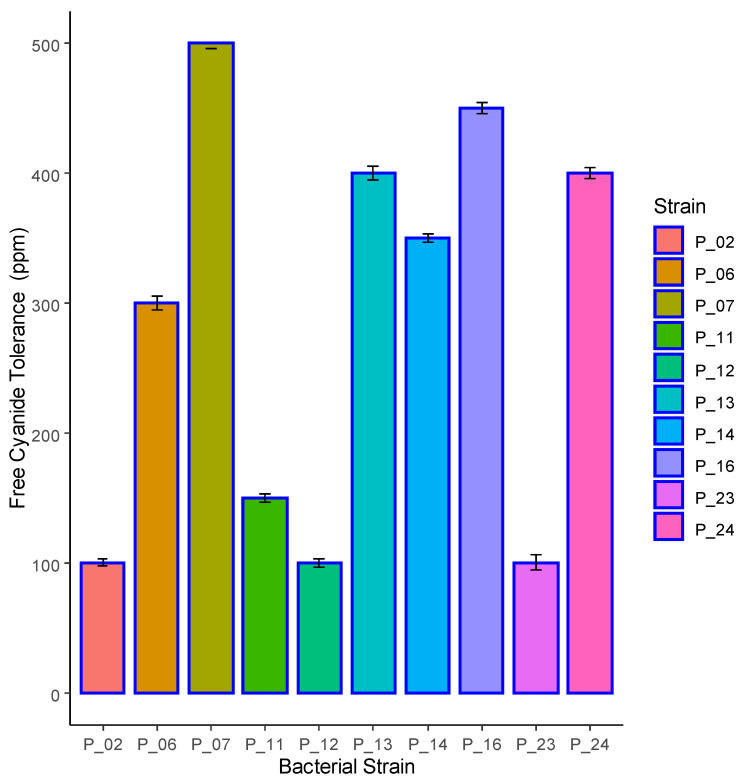
Tolerance of bacterial strains to free cyanide under alkaline conditions.

**Figure 2 microorganisms-11-02705-f002:**
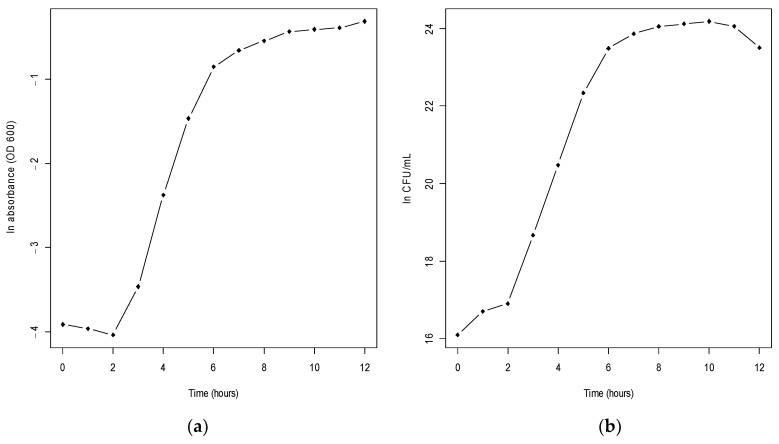
Growth kinetics. (**a**) Kinetics of the natural logarithm of absorbance (OD 600) vs. time (hours). (OD 600) vs. time (hours). (**b**) Kinetics of the natural logarithm of colony forming units (CFU/mL) vs. time (hours).

**Figure 3 microorganisms-11-02705-f003:**
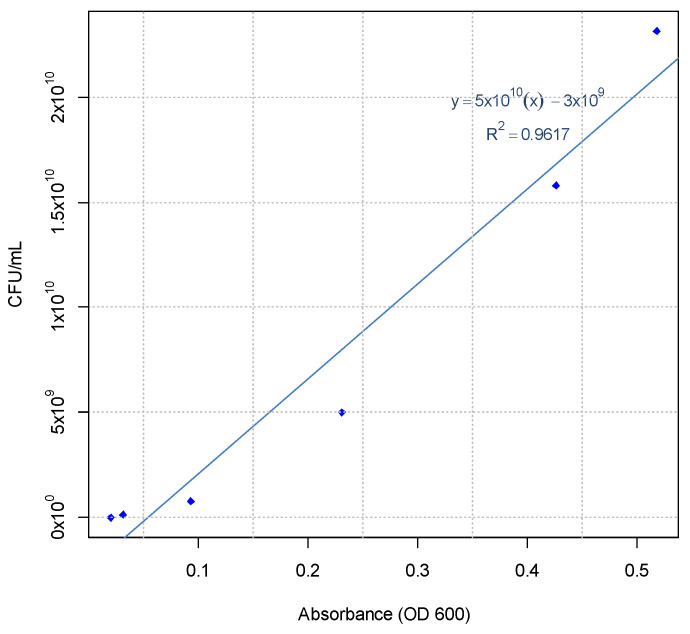
Calibration curve CFU/mL vs. absorbance (OD 600) for *Bacillus cereus* strain PBG.

**Figure 4 microorganisms-11-02705-f004:**
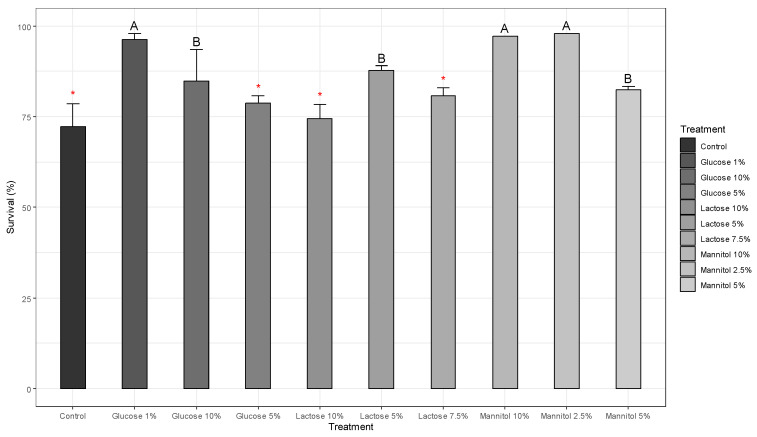
Graph of survival after lyophilization. The bars labeled with an asterisk (*) do not significantly differ from the control bar according to Dunnett’s tests. Meanwhile, bars marked with ‘A’ exhibit the highest survival rates compared to other cryoprotectants, while ‘B’ indicates lower survival rates.

**Figure 5 microorganisms-11-02705-f005:**
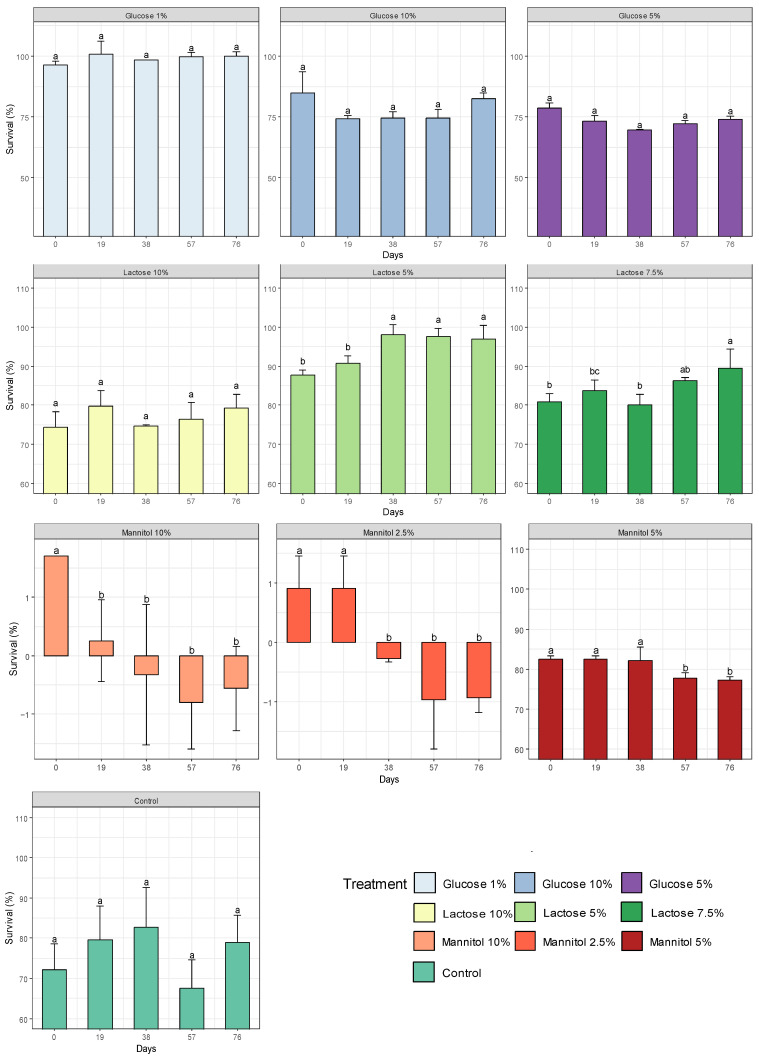
Graph of survival after lyophilization and during storage of processed sugars. Values with differing letters are considered to have significant differences.

**Figure 6 microorganisms-11-02705-f006:**
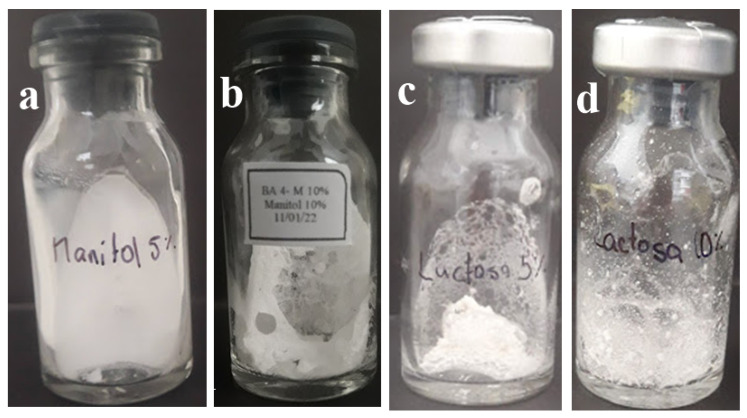
Morphology of lyophilized bacterial suspension observed. (**a**) Intact, (**b**) porous, (**c**) partial collapse, and (**d**) collapse.

**Figure 7 microorganisms-11-02705-f007:**
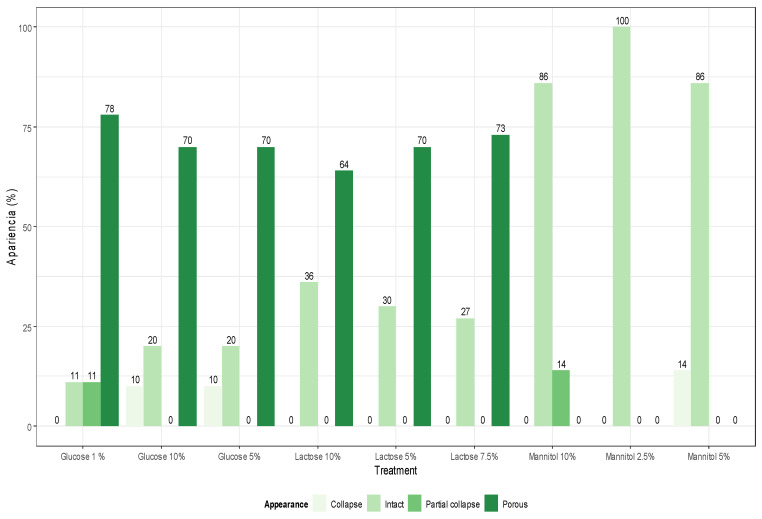
Distribution graph of cake morphology features.

**Table 1 microorganisms-11-02705-t001:** Biochemical analysis of CN^−^ Tolerance bacterial strain P_07.

Citrate	Nitrate	SIM			VP	MR	Catalase	Oxidase	O/F		
		Sulfide	Motility	Indole					Mannitol	Arabinose	Glucose
-	+	-	+	-	-	+	+	-	O/F	-/-	-/-

Voges-Proskauer Test (VP). Methyl Red Test (MR). Oxidation-Fermentation Test (O/F).

**Table 2 microorganisms-11-02705-t002:** Genes present in *Bacillus cereus* strain PBG.

Genes	Product
*nheA*	non-hemolytic enterotoxin NHE subunit A
*nheB*	non-hemolytic enterotoxin NHE subunit B
*nheC*	non-hemolytic enterotoxin NHE subunit C
*cytK*	beta-channel forming cytolysin CytK
*arsR*	arsenical resistance operon transcriptional regulator ArsR
*arsB_1*	ACR3 family arsenite efflux transporter
*arsC_1*	arsenate reductase (thioredoxin)
*cadC*	Cadmium resistance transcriptional regulatory protein CadC
*cadA_1*	cadmium-translocating P-type ATPase
*cadA_2*	cadmium-translocating P-type ATPase
*ppaC*	manganese-dependent inorganic pyrophosphatase
*dapA_1*	4-hydroxy-tetrahydrodipicolinate synthase
*serC*	3-phosphoserine/phosphohydroxythreonine transaminase
*rpsF*	30S ribosomal protein S6
*sodC*	superoxide dismutase [Cu-Zn]

**Table 3 microorganisms-11-02705-t003:** Survival values after lyophilization and during storage of processed sugars.

Treatment	N	Survival (%)				
		0 Day	19 Days	38 Days	57 Days	76 Days
Control	3	72.16 ± (6.36) a	79.60 ± (8.45) a	82.68 ± (9.92) a	67.51 ± (7.03) a	78.95 ± (6.72) a
Glucose 1%	3	96.30 ± (1.72) a	100.74 ± (5.36) a	98.32 ± (0.23) a	99.72 ± (1.71) a	99.97 ± (1.87) a
Glucose 10%	3	84.82 ± (8.72) a	74.15 ± (1.41) a	74.63 ± (2.44) a	74.49 ± (3.75) a	82.58 ± (2.13) a
Glucose 5%	3	78.64 ± (2.05) a	73.30 ± (2.25) a	69.51 ± (0.29) a	72.20 ± (1.38) a	73.96 ± (1.46) a
Lactose 10%	3	74.40 ± (4.00) a	79.79 ± (4.00) a	74.72 ± (0.26) a	76.47 ± (4.19) a	79.29 ± (3.47) a
Lactose 5%	3	87.72 ± (1.26) b	90.75 ± (1.98) b	98.05 ± (2.53) a	97.61 ± (2.04) a	97.02 ± (3.46) a
Lactose 7.5%	3	80.80 ± (2.18) c	83.78 ± (2.72) bc	80.09 ± (2.73) b	86.30 ± (0.73) ab	89.55 ± (4.81) a
Mannitol 5%	3	82.43 ± (0.93) a	82.43 ± (0.93) a	82.10 ± (3.47) a	77.75 ± (1.34) b	77.26 ± (0.86) b
* Mannitol 2.5%	3	98.02 ± (5.31) a	98.02 ± (5.31) a	80.90 ± (0.52) b	79.27 ± (1.01) b	78.71 ± (0.26) b
* Mannitol 10%	3	97.12 ± (0.03) a	81.34 ± (2.34) b	80.08 ± (1.71) b	78.83 ± (0.64) b	79.21 ± (1.14) b

In a row, values with differing letters are considered to have significant differences. (*) Survival values were transformed by Jhonson and then evaluated by ANOVA.

## Data Availability

The 16S rRNA sequences presented in this study are available in GenBank at NCBI (OR616760.1).

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
