# Peer review of "Effect of Lyoprotective Agents on the Preservation of Survival of a Bacillus cereus Strain PBG in the Freeze-Drying Process"

_microorganisms, 2023, doi:10.3390/microorganisms11112705_

Round 1

Reviewer 1 Report

Comments and Suggestions for Authors

The work of microorganisms-2677929 is devoted to the isolation of a strain of bacteria Bacillus cereus from a soil sample and the search for its lyoprotectors. The methodology is described well. Overall, the impression was that more work was done than was written. The article is written rather modestly.

To improve the material of the article, authors are recommended to do the following:

1) Justify the novelty of this study. Now the work looks like training, not scientific.

2) Add information about the beneficial properties of the selected strain. What is its value that it needs to be preserved by lyophilization? Bacillus bacteria survive spray drying very well. What valuable properties did the authors discover? How do the authors propose to use this strain?

3) Why was only one strain isolated from the soil sample? Wasn't screening done to select the best strain? If yes, what criteria were used for screening?

4) Now the authors do not seem to be firmly convinced that the isolated PBG strain really belongs to Bacillus cereus, rewrite the text regarding this point.

5) If in Figure 4 you select a scale along the ordinate axis not from 0, but from 60, then the difference in the results will be visually more noticeable.

6) Please visualize the morphology of the cake (methodology in point 2.6), provide typical photographs in Supplementary material

7) Add information about how you are going to deposit the isolated strain?

Author Response

Dear Reviewer:
We appreciate your suggestions and observations aimed at enhancing the work of our research group. Below, we are submitting the respective corrections to the manuscript.

Reviewer 2 Report

Comments and Suggestions for Authors

Manuscript entitled “Effect of Lyoprotective Agents on the Preservation of Survival of a Bacillus cereus Group Strain in the Freeze-Drying Process” submitted by Diana Farfan, Milena Carpio, Gisela Maraza, Dina Chachaque and César Cáceda, can be considered for publication in Microorganisms Journal, after a major revision.

Here is a list of my specific comments:

  1. Page 1, line 13: Replace “the survival rate” with “the survival rate of Bacillus sp. strain PBG”.
  2. Page 1, line 28: “In bioremediation processes,…”. Add here as reference the paper: doi.org/10.1016/j.envres.2023.116275, because it is relevant for this observation.
  3. Page 1, line 41: Replace “[6,10,11]; however,” with “[6,10,11]. However,”.
  4. Page 2, line 59: “The objective of this…”. The objectives of this study should be more detailed presented.
  5. Page 3, line 123: “…without melting (Figure 4).” This should be Figure 1.
  6. Page 3, 3.1. General Characteristics of Strain PBG: These observations should be supported by the experimental results.
  7. Page 4, line 160: “…concentration of 6.05x1010 CFU/mL”. Move “Figure 1” at the end of this paragraph.
  8. Page 8, 4. Discussion: In this section, all experimental results must be more detailed discussed. Provide clear explanations of these.
  9. Page 10, 5. Conclusions: This section should be detailed.

Author Response

Dear Reviewer,

We appreciate your suggestions and observations aimed at enhancing the work of our research group. Below, we are submitting the respective corrections to the manuscript.

Round 2

Reviewer 2 Report

Comments and Suggestions for Authors

All my previous remarks and comments have been considered in this new version of the manuscript. In my opinion, the revised manuscript meets the criteria and can be published as original paper in Microorganisms Journal. I would have only one observation, namely that the figures need to be rearranged. In many cases, they overlap and are hard to see.